# Simultaneous Administration of Bevacizumab with Bee-Pollen Extract-Loaded Hybrid Protein Hydrogel NPs Is a Promising Targeted Strategy against Cancer Cells

**DOI:** 10.3390/ijms24043548

**Published:** 2023-02-10

**Authors:** Nemany A. N. Hanafy, Eman Ali Bakr Eltonouby, Elsayed I. Salim, Magdy E. Mahfouz, Stefano Leporatti, Ezar H. Hafez

**Affiliations:** 1Institute of Nanoscience and Nanotechnology, Kafrelsheikh University, Kafrelsheikh 33516, Egypt; 2Department of Zoology, Research Laboratory of Molecular Carcinogenesis, Faculty of Science, Tanta University, Tanta 31527, Egypt; 3Department of Zoology, Faculty of Science, Kafrelsheikh University, Kafrelsheikh 33516, Egypt; 4CNR NANOTEC-Istituto di Nanotecnologia, Via Monteroni, 73100 Lecce, Italy

**Keywords:** bevacizumab, caspase 3, hydrogels nanoparticles, bee pollen

## Abstract

Bevacizumab (Bev) a humanized monoclonal antibody that fights vascular endothelial growth factor A (VEGF-A). It was the first specifically considered angiogenesis inhibitor and it has now become the normative first-line therapy for advanced non-small-cell lung cancer (NSCLC). In the current study, polyphenolic compounds were isolated from bee pollen (PCIBP) and encapsulated (EPCIBP) inside moieties of hybrid peptide–protein hydrogel nanoparticles in which bovine serum albumin (BSA) was combined with protamine-free sulfate and targeted with folic acid (FA). The apoptotic effects of PCIBP and its encapsulation (EPCIBP) were further investigated using A549 and MCF-7 cell lines, providing significant upregulation of Bax and caspase 3 genes and downregulation of Bcl2, HRAS, and MAPK as well. This effect was synergistically improved in combination with Bev. Our findings may contribute to the use of EPCIBP simultaneously with chemotherapy to strengthen the effectiveness and minimize the required dose.

## 1. Introduction

Chemotherapy remains the cornerstone of treatment for both non-small-cell lung cancer (NSCLC) and small-cell lung cancer (SCLC), despite the growing interest in the development of non-cytotoxic targeted medicines. Nevertheless, the resistance of chemotherapy makes advanced lung cancer difficult to successfully treat. Even those who initially react to chemotherapy frequently develop acquired resistance since some lung cancers are fundamentally resistant to it [1]. Chemotherapy has demonstrated some efficacy when administered alone to individuals at stage IV, as well as when combined with radiotherapy to treat patients with the locally progressed disease and those with early-stage NSCLC before surgery. Since chemotherapy has low efficacy and a high level of toxicity, there has been a lot of skepticism about this strategy for a long time. Only a tiny improvement in survival rates has been noted. As a widely established method of treatment for stage IIIB/IV NSCLC, there is increasing interest in the use of chemotherapy in the early stages of cancer when paired with other (local) therapies [2].

Bevacizumab (Bev), a humanized monoclonal antibody, fights vascular endothelial growth factor A (VEGF-A). It was the first specifically designed angiogenesis inhibitor [3]. Bevacizumab has improved therapeutic effectiveness since it was introduced to chemotherapy (Bev-chemotherapy), and it has now become the normative first-line therapy for advanced NSCLC [4]. Pro-angiogenic cytokines (VEGF, angiostatin-1, and follicle inhibitor) and inflammatory cytokines (interferon (IFN), IL4, and IL17) were found to be reduced in metastatic non-small-cell lung cancer treated with the EP regimen (cisplatin plus etoposide) in combination with bevacizumab compared to the conventional EP regimen alone, demonstrating its strong anti-angiogenic effects [5]. In advanced EGFR-mutant non-small-cell lung cancer, the combination of bevacizumab and the tyrosinase inhibitor erlotinib significantly increased tumor-free survival [6]. However, the benefit of Bev-chemotherapy for overall survival (OS) is still insufficient [7]. Bevacizumab use has been linked to an increase in the occurrence of arrhythmia and hypertension [8]. Reduced cardiomyocyte viability, elevated [Ca^2+^], elevated cellular mitochondrial ROS levels, mitochondrial enlargement, the collapse of the mitochondrial membrane, reduced energy metabolism, and activation of ER stress are all signs of bevacizumab-induced cardiomyocyte damage in cultured cells [9].

Polyphenolic compounds might be used to overcome the cytotoxicity of Bev-chemotherapy and strengthen its effectiveness since the immunomodulatory and vasodilatory actions of polyphenols may also contribute to a lower risk of cardiovascular disease. This reflects their cardioprotective benefits [10]. Meanwhile, flavonoids display their anticancer effect by blocking protein kinases and pro-oxidant enzymes, controlling the metabolism of carcinogens, reducing drug resistance, suppressing angiogenesis, and triggering apoptosis and cell cycle arrest. Additionally, flavonoid-rich meals and drinks can lower blood pressure, enhance endothelial function, and increase nitric oxide bioavailability to avoid cardiovascular disease [11].

One of the natural products with the highest concentrations of proteins, polysaccharides, polyphenols, lipids, minerals, and vitamins is bee pollen. It has a substantial positive impact on health and medicine and offers defense against a wide range of illnesses, including diabetes, cancer, infectious, and cardiovascular disorders [12]. Unfortunately, the anticancer potential of dietary flavonoids is insufficient due to their poor solubility, quick metabolism, and absorption [13]. Recently, hydrogel NPs made of bovine serum albumin (BSA) have provided potential applications as drug delivery systems due to their low cost, their abundant chemical interactions, their biodegradability, and their mucoadhesive properties, and about 76% similarity with the molecular structure of human serum albumin [14,15,16,17]. In the current study, protamine as a cationic peptide is used to bind albumin through ionic interaction forming a stable electrostatic assembly. Protamine is a non-toxic peptide that can penetrate the cell membrane and also facilitate intestinal absorption. Additionally, folic-acid-conjugated protamine was used to improve the efficacy of tumor-targeted delivery [18].

## 2. Results

### 2.1. Flavonoid Composition of PCIBP and EPCIBP by HPLC

The amount of each flavonoid in the PCIBP and EPCIBP are displayed in Table 1 and Figure 1. According to PCIBP, vanillin (0.082 0.001 g/mL) had a lower concentration than quercetin (6.43 0.18 g/mL). Gallic acid, cinnamic acid, syringic acid, catechin, taxifolin, ferulic acid, pyro catechol, caffeic acid, naringenin, coumaric acid, rutin, chlorogenic acid, and methyl gallate were the components of the other contents of flavonoids.

Similarly, the flavonoids discovered in the EPCIBP demonstrate that quercetin (5.230.01 g/mL) made up the main concentration, whereas methyl gallate (0.0550.001 g/mL) was the minor component. Cinnamic acid, gallic acid, taxifolin, syringic acid, catechin, naringenin, caffeic acid, pyro catechol, rutin, ferulic acid, coumaric acid, chlorogenic acid, and vanillin made up the remaining ingredients.

While encapsulation of bee pollen extract resulted in minimizing the concentrations of its polyphenolic content, gallic acid and taxifolin were still major components. This result could be explained by the fact that polyphenolic compounds (crude) may provide various interactions during their encapsulation depending on their chemical reaction with certain types of polymers or materials since polyphenolic compounds mainly contain carboxylic groups and hydroxyl groups that can react either by ester bond or amide bond. Based on their interaction and location inside capsules, they may be isolated and released from their location simply or they can be attached strongly and stay in the inner face of polymer moieties. Therefore, isolating polyphenolic compounds from capsules does not reveal their real concentrations [19,20].

### 2.2. Characterization of EPCIBP

Encapsulating bee pollen extract aims to preserve its bioactive compounds, allowing for improvement in their biological activity as well as ensuring their controlled release.

SEM imagery of EPCIBP exhibited a smooth surface with a nearly spherical shape (Figure 2A,B). The histogram average size was estimated at 116 nm (Figure 2C). The formation of polyphenolic compounds inserted into BSA was assembled mainly in a 3D structure owing to the presence of several hydroxyl groups in polyphenols’ structure that could interact with amino groups of BSA. The reaction of polyphenolic compounds with albumin led to the formation of micro/nanopores distributed inside a matrix complex (Figure 2D–F). Meanwhile, micro/nanopores were absent from the matrix of free capsules obtained by the reaction of BSA with PRM (Figure 2G,H). This result indicates that polyphenolic compounds (crude) can be reacted with a protein either by reversible non-covalent interactions organized by hydrophobic bridging, hydrogen bonding, van der Waals force, and ion interaction or by irreversible covalent interaction formed through covalent bonding [20,21].

The mean diameter of free capsules, as determined by dynamic light scattering (DLS) of the zeta sizer, was 51 nm, indicating good distribution and heterogenicity (Figure 3A). Meanwhile, the diameter of EPCIBP was determined to be 138.2 nm, and the polydispersity index (PDI) was 0.3 with good distribution (Figure 3B).

Zeta potential measurements of EPCIBP were detected at −17.6 2 mV (Figure 3C), while the net charge of free capsules was obtained at −8.8 1.5 mV (Figure 3D). These results indicate that polyphenolic compounds can change the final charge after being integrated into BSA moieties (Figure 3C) and can provide strong physical stability as a result of nanoparticle electrostatic repulsion.

### 2.3. Determination of the Loading and Targeting Capacity of the EPCIBP Conjugated by FA

In the current study, the amount of FA-conjugated EPCIBP was calculated from a standard curve of 1 mg/mL pure FA (R^2^ = 0.9983). After the encapsulation process, the supernatant was collected by centrifugation, and the concentration of non-conjugated FA was estimated. The percentage of encapsulation efficiency of FA was detected as 98 ± 0.23% (Figure 4A–C). The result showed significant conjugation of FA with protamine because of the amino contents of protamine and the good carboxylic activation of FA. A similar result was obtained in our previous publications, confirming that FA dissolved by NaOH can conjugate strongly through amide bond interaction with opposite polymers such as chitosan and protamine and can form ester bonds with polyethylene glycol terminal OH [22,23,24,25].

To understand the modification of chemical bands during conjugation, FA, PRM, and FA-conjugated PRM were studied using FTIR. The reaction of the FA–PRM complex was confirmed by the appearance of a peak at 1377 cm^−1^ belonging to the C = N bond on FA. In the same way, the characteristic peak (1676 cm^−1^) representing −COOH groups on FA was shifted to 1647 cm^−1^, which corresponded to amino-carboxylic bond formation between FA and PRM, also indicating the successful attachment of FA with PRM (Figure 4D).

The targeting capacity of EPCIBP accumulated in the prenuclear region of A549 cells was investigated by fluorescence microscopy. R6G-labeled EPCIBP (Figure 5A) was successfully localized inside the cytoplasm, as demonstrated by the fluorescence intensity of TRITC color (red) located in the perinuclear region (Figure 5B,C). Fluorescence images prove that EPCIBPs were readily collected in cellular compartments, leading to increased drug capacity and effectiveness. The corrected total cell fluorescence (CTCF) was used to quantify the fluorescence intensity in each cell compared to other cells. To evaluate the level of cellular fluorescence emission, the fluorescence of the background was used to normalize the intensity of the interesting cells. In the current study, CTCF was detected at 24 and 48 h as 3232 ± 1.2 and 7765 ± 2, respectively. This indicates that the accumulation of nanoparticles in cancer cells depends mainly on their size, functionalization, time of their exposure, type of cancer cells, and their ligand-targeted delivery [26].

### 2.4. Antiproliferative Effect In Vitro Using A549 Cells

An MTT assay was used in the current study to evaluate the potential antiproliferative effect of PCIBP, EPCIBP, Bev., and their combination.

Here, the percentage of cell inhibition was measured after their incubation to serial concentrations of PCIBP (20, 40, 60, 80, and 100 µg/mL). The results showed significant inhibition in growth of A549 after 24 h of incubation (7.15 ± 0.05%, 18.9 ± 0.01%, 54.8 ± 0.03%, 58.7 ± 0.05%, and 42.6 ± 0.02%, respectively) and after 48 h of incubation (38.8 ± 0.01%, 48.2 ± 0.02%, 65.5 ± 0.09%, 70.9 ± 0.04%, and 84.8 ± 0.01%, respectively) (Figure 6A).

Likewise, at serial concentrations of (62.5, 125, 250, 500, and 1000 µg/mL), the percentage of cell inhibition of A549 cell lines exposed to EPCIBP for 24 h of incubation was (3.5 ± 0.02%, 12 ± 0.03%, 36 ± 0.02%, 61.9 ± 0.11, or 75 ± 0.09%, respectively) and for 48 h incubation was (8 ± 0.17%, 43 ± 0.02%, 56.5 ± 0.1%, 89.8 ± 0.03, or 96.4 ± 0.005%, respectively) (Figure 6B).

In the same way, the percentage of cell inhibition was measured to A549 after their incubation with different concentrations (10, 20, 40, 60, and 80 80 µg/mL) of Bev. The results showed significant inhibition in growth of A549 after 24 h of incubation (83.0 ± 0.01%, 10.0 ± 0.01%, 16.6 ± 0.01%, 23.9 ± 0.01%, and 42.3 ± 0.03%, respectively) and after 48 h of incubation (7.0 ± 0.05%, 17.0 ± 0.05%, 23.7 ± 0.04%, 32.2 ± 0.05%, and 65 ± 0.01%, respectively) (Figure 6C). The half-maximal inhibitory concentration (IC50) values for PCIBP, EPCIBP, and Bev. were detected after 24 of incubation (31.73, 214.3, and 62.2 µg/mL, respectively) and after 48 h of incubation (17.6, and 132.4 and 26.1 µg/mL, respectively).

### 2.5. Antiproliferative Effect In Vitro Using MCF-7 Cells

MCF-7 is made up of human adenocarcinoma cells, characterized by their overexpression of glucocorticoid, progesterone, and estrogen. MCF-7 was used in the current study because it maintains characteristics similar to those present in mammary epithelium. Based on the cytotoxicity assay, the growth of MCF-7 was inhibited significantly after exposure to serial concentrations of PCIBP, EPCIBP, and Bev. (Figure 7A–C). The results showed that PCIBP inhibits the percentage of MCF-7 cells after 24 h of incubation (24.4 ± 0.01%, 30.1 ± 0.01%, 33.5 ± 0.02%, 34.6 ± 0.01%, and 35.2 ± 0.02%) and after 48 h of incubation (35.6 ± 0.03%, 50.4 ± 0.01%, 54.5 ± 0.04%, 58 ± 0.03%, and 69 ± 0.02%).

Similarly, EPCIBP provides the ability to inhibit the growth of MCF-7. The percentage of cell inhibition was significantly modified with serial concentrations of EPCIBP (62.5, 125, 250, 500, and 1000 µg/mL) after 24 incubation (30.4 ± 0.17%, 33.6 ± 0.03%, 34.6 ± 0.04%, 54.7 ± 0.05, and 55.9 ± 0.01%, respectively) and after incubation for 48 h (27.6 ± 0.03%, 41.5 ± 0.005%, 44.6 ± 0.01%, 61.4 ± 0.03, and 69.8 ± 0.02%, respectively).

In the same way, the cytotoxicity of Bev. (10, 20, 40, 60, and 80 µg/mL) was studied after its incubation with MCF-7 for 24 h (11.3 ± 0.01%, 20.9 ± 0.009%, 21.6 ± 0.01%, 27.6 ± 0.03%, and 36.5 ± 0.03%, respectively) and for 48 h (42.5 ± 0.01%, 43.5 ± 0.004%, 48.4 ± 0.01%, 49.4 ± 0.007%, 60.6 ± 0.02%, respectively).

The IC50 values were calculated for PCIBP, EPCIBP, and Bev. after 24 and 48 h. The cytotoxic activity of PCIBP, EPCIBP, and Bev. was detected after 24 h of incubation (IC50 > 27.89 µg/mL, IC50: 125.6 µg/mL, and IC50: 35.28 µg/mL, respectively) and after 48 h (IC50 > 13.43 µg/mL, IC50: 92.83 µg/mL, and IC50: 6.32 µg/mL, respectively).

Normal Vero cells were used to evaluate the cytotoxicity of PCIBP, EPCIBP, and Bev. Vero cells are a lineage of cultured cells used in the laboratory as normal cells. The ‘Vero’ lineage was isolated from kidney epithelial cells extracted from an African green monkey. The result was measured through spectrophotometry using an MTT assay after 24 h and 48 h (Figure 8A–C). The growth of Vero cells showed gradual inhibition after their exposure to PCIBP for 24 h of incubation (7.5 ± 0.02%, 14.6 ± 0.03%, 14.8 ± 0.01%, 17.4 ± 0.003%, and 35.2 ± 0.02%) and after 48 h of incubation (28.5 ± 0.03%, 34.8 ± 0.007%, 43.3 ± 0.01%, 52.0 ± 0.01%, and 62.3 ± 0.01%).

EPCIBP showed low toxicity after incubation with Vero cells for 24 h and 48 h. The percentage of cell inhibition of serial concentrations of EPCIBP (62.5, 125, 250, 500, and 1000 µg/mL) showed low toxicity after 24 h (4.6 ± 0.02%, 2.6 ± 0.001%, 13.4 ± 0.04%, 15.3 ± 0.04, and 16.7 ± 0.006%, respectively) and after 48 h (20.2 ± 0.02%, 29.8 ± 0.04%, 32 ± 0.04%, 35 ± 0.04, and 37 ± 0.003%, respectively).

Meanwhile, Vero cells showed gradual toxicity with serial concentrations of Bev. (10, 20, 40, 60, and 80 µg/mL) after 24 h of incubation (1.7 ± 0.002%, 2.7 ± 0.004%, 8.4 ± 0.01%, 10.1 ± 0.01%, and 25.7 ± 0.02%, respectively) and after 48 h of incubation (8.7 ± 0.08%, 15.3 ± 0.01%, 15.4 ± 0.01%, 36.3 ± 0.03%, and 43.0 ± 0.009%, respectively).

### 2.6. Cytotoxic Synergism of Bev. with EPCIBP in Non-Small Lung Cancer Cells

When Bev. was combined with EPCIBP at IC50 doses of 1:1, 1:4, 4:1, 1:9, and 9:1 and then administered by A549 cells for 48 h, combination index (CI) plots and standard isobolograms were used to evaluate the kind of drug–drug interaction. EPCIBP and Bev. demonstrated a definite synergistic impact (1) on A549 cells (Table 2). The 4:1 and 1:1 combination ratios (CI = 0.01305 and 0.01771, respectively) were the doses with the most common synergistic effect, followed by 1:4 (CI = 0.08764), 9:1 (CI = 0.28923), and 1:9 (CI = 0.40409) (Table 2 and Figure 9). According to statistical analysis, the results from the combinations 1:1 and 4:1 did not show a significant difference (*p* < 0.987), while the other combinations created significant differences (*p* < 0.0001).

### 2.7. Expression of Apoptotic Genes in Untreated and Treated A549 Cell Lines

The mean normalized Bax mRNA gene expression was significantly elevated in all treatments (PCIBP, EPCIBP, Bev., or their combination) as follows: 2.31 ± 0.01, *p* < 0.0001; 1.65 ± 0.02, *p* < 0.05; 2.55 ± 0.02, *p*< 0.0001; and 2.89 ± 0.01, *p* < 0.0001, respectively (Figure 10A).

Using the same technique, the expression of the Bcl2 mRNA gene was found to be downregulated in all treatments compared to the untreated cells, as shown in Figure 10B. The treatments—PCIBP, EPCIBP, Bev., or their combination—produced the following results: 0.55 ± 0.03, *p* < 0.001; 0.21 ± 0.01, 0.09 ± 0.005, and 0.19 ± 0.01; *p* < 0.00001, respectively.

The expression of the caspase 3 mRNA gene was significantly increased in all treatments (PCIBP, EPCIBP, Bev., or their combination) when compared to the untreated cells. The results showed a significant increase in the level of caspase 3 mRNA (1.79 ± 0.01, *p* < 0.05, 1.92 ± 0.01, *p* < 0.001, 2.69 ± 0.04, *p* < 0.00001, and 1.28 ± 0.01, respectively) (Figure 10C).

### 2.8. Expression of the HRAS mRNA Gene in Untreated and Treated A549 Cells

The expression of HRAS mRNA levels in A549 cells treated or untreated with IC50 of PCIBP, EPCIBP, Bev., or their combination was investigated after 48 h (Figure 11A). In all groups treated with PCIBP, EPCIBP, Bev., or their combination, the average normalized expression (RQ) of the HRAS gene was discovered to be downregulated as follows: 0.18 ± 0.01, 0.06 ± 0.001, 0.1 ± 0.005, and 0.22 ± 0.01, respectively (*p* < 0.00001).

### 2.9. Expression of MAPK mRNA in Untreated and Treated A549 Cells

In contrast to the untreated cells, all treatments—aside from PCIBP—significantly raised the MAPK gene’s mean normalized expression (RQ), except for PCIBP treatment, which decreased it by the following amounts: 1.6 ± 0.08, *p* < 0.00001, 0.73 ± 0.01, *p* < 0.5, 0.14 ± 0.005, *p* < 0.00001, and 0.77 ± 0.005, respectively (Figure 11B).

### 2.10. Cancer Bio-Image

A mixture of acridine orange and ethidium bromide was used to visualize morphological alterations of cancer cells under fluorescence microscopy after their exposure to chemotherapies. Ethidium bromide could penetrate cell membranes after their integration and then was attached to DNA.

In the current study, non-small lung cancer cells (A549 cells), breast cancer cells (MCF-7), and Vero cells as normal controls were treated by PCIBP, EPCIBP, Bev., and a combination of EPCIBP with Bev. The result showed nuclear condensation and fragmentation. In addition, A549 cells were mostly swelled in case of using the combination (Figure 12).

Likewise, MCF-7 exposed to PCIBP, EPCIBP, Bev., and their combination exhibited apoptotic features including nuclear activation with the presence of early apoptotic stages. The results showed that EPCIBP has a potentially greater effect on MCF-7 compared to PCIBP. Meanwhile, the combination showed strong nuclear activation and cytoplasmic degradation (Figure 13A–D).

The potential cytotoxicity of PCIBP, EPCIBP, Bev., and their combination was also evaluated using normal Vero cells. The results showed that PCIBP, Bev., and their combination have a cytotoxic effect on normal Vero cells and they can induce apoptotic features, while MCF-7 exposed to EPCIBP exhibited a normal morphology (Figure 13E–H).

## 3. Discussion

The application of bee pollen in the pharmacological industry has been extensively studied in the antibacterial, antiviral, and anti-inflammatory fields [27]. The current study focused on determining the anti-cancer properties of polyphenolic compounds isolated from Egyptian bee pollen and encapsulated in hybrid protein–polymer hydrogel NPs and then studying their effect in combination with bevacizumab chemotherapy.

HPLC results quantified the polyphenols and flavonoids isolated from Egyptian pee pollen by alcoholic methods (Figure 1 and Table 1), and they were demonstrated as follows: cinnamic acid, syringic acid, catechin, gallic acid, taxifolin, ferulic acid, pyro catechol, naringenin, coumaric acid, rutin, chlorogenic acid, methyl gallate, and caffeic acid. 

This result is in agreement with [28,29], who isolated numerous polyphenolic compounds from bee pollen. However, the concentrations of these molecules differed between the encapsulation and extract. This result could be explained by the fact that polyphenolic compounds (crude) may provide various interactions during their encapsulation depending on their chemical reactions with certain types of polymers or materials. Polyphenolic compounds mainly contain carboxylic groups and hydroxyl groups that can react either by ester bond or amide bond. Based on their interaction and location inside capsules, they may be isolated and released from their location simply, or they can be attached strongly and stay in the inner face of polymer moieties. Therefore, isolating polyphenolic compound from capsules does not reveal their real concentration [30]. This finding was supported by [31,32].

To improve the application of polyphenolic compounds in the pharmaceutical industry, and to enhance their antioxidant efficiency, bee pollen extract was integrated into moieties of bovine serum albumin, and the mixture was coated by FA-PRM. Along the same lines, the extract of bee pollen was encapsulated previously in chitosan moieties to improve its antioxidant and antibacterial activities [33].

Here, SEM images revealed an assembled structure in a spherical shape with many distributed micro/nanopores in the matrix of the structure. EPCIBP exhibited a homogenous distribution of negatively charged particles with a diameter of 116 nm. Folic-acid-conjugated PRM was characterized by the presence of its two main peaks at 285 nm and 365 nm in the spectrum of EPCIBP (Figure 2, Figure 3 and Figure 4) [34].

The potential cytotoxicity of PCIBP, EPCIBP, Bev., or their combination was investigated by using non-small lung cancer (A549 cell lines), breast cancer (MCF-7 cell lines), and normal Vero cells. The results highlighted the significant potency of EPCIBP in cancer treatment in combination with chemotherapies or in separate administration (Figure 6, Figure 7 and Figure 8).

To understand the mechanism by which PCIBP, EPCIBP, Bev., or their combination can affect non-small lung cancer (A549 cell line) compared to untreated cells, MAPK, RAS genes (HRAS), and apoptotic gene expression (Bcl 2, Bax, and caspase 3) were evaluated.

The results confirmed that EPCIBP administrated separately or simultaneously with Bev. can induce significant downregulation of MAPK. The result is constant with [35] who reported that flavonoids can regulate the MAPK signaling pathways.

The significant inhibition of the HRAS gene was explored in the current study using PCIBP, EPCIBP, Bev., or their combination. The result showed that mRNA expression of the HARS gene was inactivated considerably, as follows: 0.18 ± 0.01, 0.06 ± 0.001, 0.1 ± 0.005, and 0.22 ± 0.01, respectively (*p* < 0.00001). This evidence was confirmed by [36], who reported that polyphenolic compounds and flavonoids can affect the Ras family in lung cancer. Similarly, polyphenols dramatically decreased the expression of proliferation-associated genes such as HRAS, c-myc, and cyclin D1 [37].

In seeking to improve our understanding of the molecular mechanistic basis for the chemopreventative properties of EPCIBP, we demonstrated that EPCIBP inhibited cell proliferation and induced apoptosis in A549 lung cancer cells. The upregulation of pro-apoptotic genes such as Bax and caspase 3, and the downregulation of an anti-apoptotic gene such as Bcl-2 were evaluated while PCIBP, EPCIBP, Bev., or their combination induced growth inhibition and activation of apoptotic stages [38]. The results showed a significant increase in the levels of Bax and caspase 3 mRNA after treatment of A549 cell lines with PCIBP, EPCIBP, Bev., or their combination, while there was significant inhibition of the level of Bcl 2 after the treatments (Figure 10).

## 4. Materials and Methods

### 4.1. Extraction of a Polyphenolic Compound from Bee Pollen

To begin with, 3 g of powdered bee pollen was stored in a flask with a mixture of 5 mL DMSO and 45 mL of 96% ethanol. The mixture was then agitated for 1 h at 55 °C and 200 rpm using a magnetic stirrer. The supernatant was removed by centrifugation for 10 min at 6000 rpm [34]. After that, the supernatant was kept at 4 °C until it was used.

### 4.2. Synthesis of Encapsulation of Polyphenolic Compound Isolated from Bee Pollen (EPCIBP)

Bee pollen extract (20 mL) and bovine serum albumin (BSA) (50 mg/50 mL) were mixed for 30 min at room temperature while being stirred with a magnetic starrier [35]. The assembly was further functionalized by folic acid conjugated protamine (PRM) (50 mg/100 mL) and stirred magnetically for 15 min. The BPENP was then dialyzed for 24 h in a dialysis bag against distilled water (retained molecular weight: 10,000–14,000 Da) [36]. The distilled water was changed every 6 h to remove unreacted components and then stored at −20 °C for lyophilization [32].

### 4.3. Characterization of PCIBP and EPCIBP

#### 4.3.1. HPLC for PCIBP and EPCIBP

The EPCIBP or PCIBP was centrifuged at 14,000 rpm for 10 min after being individually sonicated for 15 min with absolute ethanol at 5 amplitudes for HPLC analysis. HPLC (Agilent Technologies 1200 Infinity series, Santa Clara, CA, USA) was used to identify the encapsulated flavonoids in the supernatant [28,29].

#### 4.3.2. UV Spectrophotometer of PCIBP and EPCIBP

Using a UV spectrophotometer (Thermo Scientific, Mercers Row, Cambridge, UK) set to 280 nm, the absorption bands of BPE or BPENP were identified. The UV-Vis spectra were recorded at room temperature on a TU-1901 spectrophotometer built by Puxi Analytic Instrument Ltd. (Beijing, China) with 1 cm quartz cells. The wavelength was measured between 340 and 200 nm [39].

#### 4.3.3. Nanosize and Zeta Potential Measurements of PCIBP and EPCIBP

A Zeta Sizer Nano ZSP from Beckman Analytical Technologies was used to analyze the hydrodynamic size of nanoparticles and surface charge, or the zeta potential (v7.13). The zeta potential was observed using the M3 phase analysis light-scattering mode after NPs were diluted in a 1:2 ratio (NPs: H_2_O) to quantify the particle size [40].

#### 4.3.4. Scanning Electron Microscopy (SEM) for EPCIBP

To assess the surface morphology of the nanoparticles, a SEM examination was used. The samples were added to an aluminum stub covered in double-sided sticky tape. Through JEOL expository SEM, photomicrographs were produced of the samples, which had been randomly sputter-coated with gold [41].

#### 4.3.5. Characterization of FA

The concentration of FA attached successfully using EPCIBP was quantitatively analyzed using a UV-Vis spectrophotometer. Firstly, the calibration curve of standard FA in 1 mg/mL NaOH was prepared with a concentration range of 25–150 ug/mL (y = 0.0149x − 0.0247; R^2^ = 0.9983). After the encapsulation process, the supernatant was used to detect the loading capacity of the FA conjugation using the following equations: Concentration of non-conjugated FA (supernatant) (µg/mL) = (slope × absorbance) ± intercept
Encapsulation efficiency of FA (%) = Cconc. of total FA − conc. of free FA (supernatant)\conc. of total FA × 100

#### 4.3.6. FTIR Experiment

FTIR experiments were carried out using the JASCO Fourier Transform Infrared Spectrometer (JASCO, Tokyo, Japan, model no. AUP1200343) to detect the surface molecular structures in the scale bar of 400–4000 cm^−1^ via the KBr pellet method. Folic acid, protamine, and FA-PRM were used to identify chemical band modification, three scans were recorded on different regions of the samples, and representative spectra were analyzed.

### 4.4. Cellular Uptake and Targeting Capacity

A549 cells (10^6^) were planted on the surface of a sterilized coverslip and placed on the bottom of 6-multiwell plates. After 24 h of growth, 50 μg/mL (rhodamine-EPCIBP) was added to each well and incubated for 24 h and 48 h in a humidified atmosphere of 37 °C and 5% CO_2_. A549 cell lines were cleaned with phosphate buffer saline and then fixed with 4% paraformaldehyde. Cells were further washed by PBS, pH 7.2. Cellular uptake was investigated after 24 h and 48 h by red (TRITC) channels of fluorescence microscopy, and then images were taken by a digital camera. Targeting capacity was established by the intensity of collected NPs in the perinuclear region of the cytoplasm

The fluorescence intensity was measured using the Image J program (http://rsbweb.nih.gov/ij/download.html) according to Hanafy et al. From the Analyze menu in the Image J program, “set measurements” was chosen and the area integrated intensity and mean grey values were activated. After that, the selected image was transferred into grayscale production, the interesting cell was distinguished by the circle tool, and the intensity was measured from the analysis menu. The corrected total cell fluorescence (CTCF) was calculated [42].
CTCF = (mean fluorescence of background readings *×* area of a selected cell) − integrated density.

### 4.5. MTT Analysis/Cell Proliferation Assay

The lung cancer (A549) cell line, breast cancer (MCF-7) cell line, and Vero cells from monkey kidney were maintained in a standard medium made up of DMEM with 10% (*v*/*v*) fetal bovine serum and 1% (*v*/*v*) penicillin/streptomycin, which was subsequently changed with new DMEM−10% FBS after an initial incubation at 37 °C and 5% CO_2_. At a density of 1 × 10^4^, the cells were seeded in 96-multiwell culture plates under normal conditions until exponential growth occurred. Different concentrations of PCIBP, EPCIBP, Bev., or a mix of EPCIBP and Bev. were used to treat the cells. After the incubation period (24 or 48 h), the medium was taken out, 5 mg/mL of MTT was added, and the mixture was incubated for 4 h. The formazan crystals were dissolved in 100 mL of acidified isopropanol, and an ELISA microplate reader was used to read the results at 570 nm (Bio-RAD microplate reader, Japan). Three duplicates of each concentration were performed. Cell viability (%) = (Abs. s/Abs. c) ∗ 100, where (Abs) of sample and (Abs) of control are, respectively, the absorbances of the cells treated with samples and those not incubated with samples [43]. The experiment’s concept is authorized by Tanta University’s Zoology Department’s Institutional Animal Care and Use Committee (IACUC) (IACUC-SCI-TU-0129).

#### 4.5.1. Combination Therapy Bioassay

EPCIBP (IC50) and Bev. were combined in non-constant ratios and administered to A549 cells at ratios of 1:1, 1:4, 4:1, 1:9, and 9:1 after MTT test calculations of the IC50 levels for each treatment compound were completed. Bev. and EPCIBP IC50 fractions were used to create all combinations. The best treatment combination for the cell line was that with the lowest Bev. dosage and combination index compared to single-drug therapy (CI). All treatments were cultured for 48 h in a CO_2_ incubator, after which they were trypsinized, collected, and immediately subjected to molecular examinations.

#### 4.5.2. Drug Combination Analysis

The combination index (CI) was developed using data on the effective dose threshold (IC50) from the MTT assays based on concentration–effect curves produced as a plot of the fraction of unaffected cells vs. drug concentration [19], using CompuSyn software. This work was completed to investigate the synergism between EPCIBP and Bev. against A549 cells (PD Science, LLC, Lakefront Dr Earth City, MO, USA). The CI values signify an impact that is additive when they are equal to 1, antagonistic when they are larger than 1, or synergistic when they are less than 1.

### 4.6. RNA Isolation and Real-Time Quantitative PCR

Using a commercial kit, total RNA was extracted after obtaining the IC50 values from untreated or treated cells (QIAGEN, Analytik Jena Biometra AG, Germany). A Nanodrop 2000 (Thermo Scientific, San Jose, CA, USA) was used to measure the total RNA in each sample. Using Sensiscript Reverse Transcriptase, RNA samples were reverse transcribed into cDNA (QIAGEN, Jena, Germany). The following primers were used for RT-PCR. HRAS: forward, 5′-CTCGCAGCTATGGCATCC-3′, reverse, 5′-CAACGTGTGCCCTCACAG-3′; MAPK: forward, 5′-TCAAGCCTTCCAACCTC-3′, reverse, 5-GCAGCCCACAGACCAAA-3′; Bax: forward, 5′-TCCCCCCGAGAGGTCTTTT-3′, reverse, 5′-CGGCCCCAGTTGAAGTTG-3′; Bcl2: forward, 5′-TTGGCCCCCGTTGCTT-3′, reverse, 5′-CGGTTATCGTACCCCGTTCTC-3′; caspase 3: forward, 5′-TCCTCCTTTGCCAACACACA-3′, reverse, 5′-TGACCCATTTCATCACCCACT-3′; B-actin: forward, 5′-CTGTCCCTGTATGCCTCTG-3′, reverse, 5′-ATGTCACGCACGATTTCC-3′. RT-PCR was carried out in two steps using the SYBR GREEN PCR master mix (QuantiTect^®^ SYBR^®^ Green I kit for quantitative RT-PCR). The amount of mRNA was calculated by the comparative CT method, which depends on the ratio of the number of target genes to reference gene B-actin [44].

### 4.7. Cancer Bio-Image

A549 cells were cultured in 24-multiwell plates (1 × 10^4^ cells/well). The cells were then grown in 500 μL DMEM high-glucose (405 g/L) supplemented with 5% L-glutamine, 10% fetal bovine serum, and 5% penicillin/streptomycin in a humidified atmosphere with 5% CO_2_ at 37 °C. After 24 h, 100 µg of PCIBP, EPCIBP, Bev., and combination were added and incubated for 24 h in a humidified atmosphere of 37 °C, 5% CO_2_. A549 cells were fixed with 4% paraformaldehyde before being washed with PBS, pH 7.2. Cells were stained separately using AO/EB and DAPI (nuclear stain) for 30 min and then washed twice. Images were captured by fluorescence microscopy.

### 4.8. Statistical Analysis

Group data expressed as means ± S.D. were analyzed using the two-tailed *t*-test or ANOVA analyses, while the data expressed as percentages were analyzed with the chi-squared (X2) using GraphPad Prism, version 8.2.0 (USA). *p* ≤ 0.05 was considered significant.

## 5. Conclusions

The current study has reported that EPCIBP can be administrated simultaneously with chemotherapies to potentiate their effect and minimize their required dose. The polyphenolic components of bee pollen strengthen their potential chemo-preventive and protective agents and can be used as a dietary supplement during the chemotherapeutic course. The loading of bevacizumab and polyphenolic compounds in individual NPs used as targeted therapy represents a new area of research that should be focused on in the future.

## Figures and Tables

**Figure 1 ijms-24-03548-f001:**
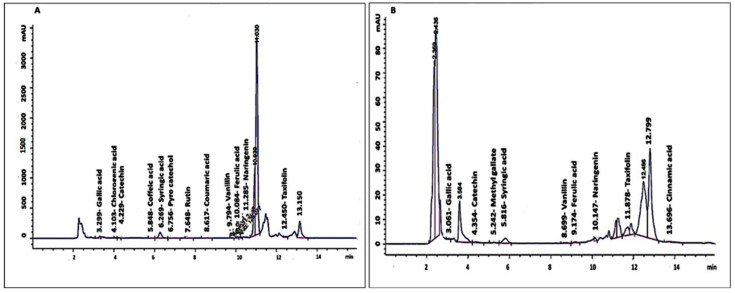
HPLC chromatography results show peaks of flavonoid contents of isolated PCIBP and EPCIBP. (**A**) PCIBP; (**B**) EPCIBP.

**Figure 2 ijms-24-03548-f002:**
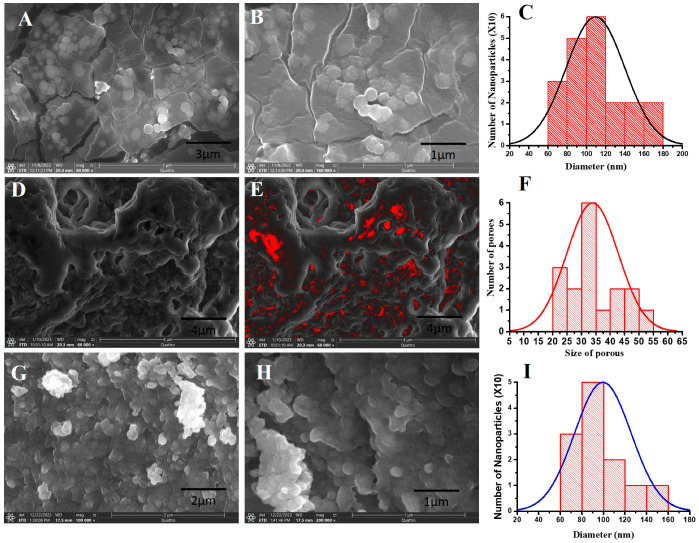
SEM image showing the morphology of EPCIBP. (**A**) SEM image of EPCIBP showing nanoparticles with spherical shapes; (**B**) magnification image of EPCIBP; (**C**) quantification analysis of diameter of nanoparticles; (**D**) SEM image of EPCIBP showing micro/nanopores; (**E**) grayscale image with thresholding observing the micro/nanopores as red color; (**F**) quantification analysis of micro/nanopores distributed in the matrix; (**G**) SEM image of free capsules showing nanoparticles with spherical shape; (**H**) magnification image of free capsules; (**I**) quantification analysis of diameter of nanoparticles.

**Figure 3 ijms-24-03548-f003:**
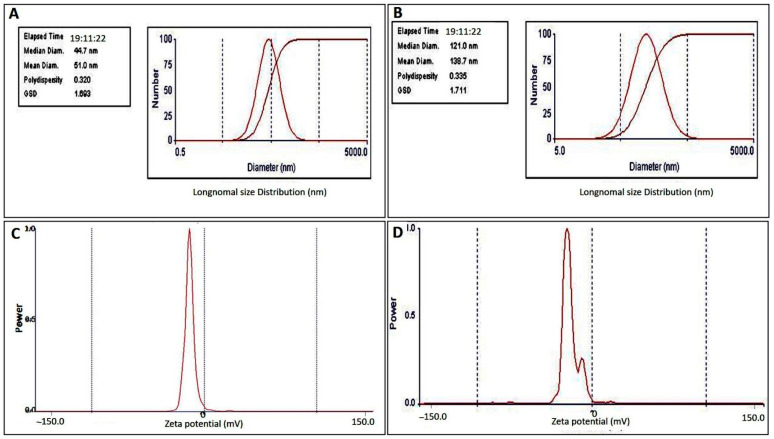
DLS and zeta potential measurement of free capsules and EPCIBP. (**A**) DLS of free capsules; (**B**) DLS of EPCIBP; (**C**) zeta potential of free capsules; (**D**) zeta potential of EPCIBP.

**Figure 4 ijms-24-03548-f004:**
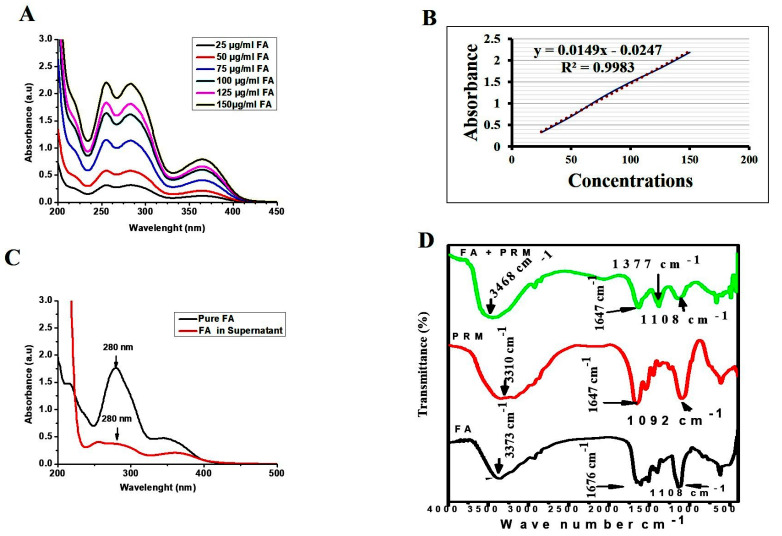
(**A**) Standard curve of folic acid. (**B**) Graph of folic acid calibrating curve (red is trendline and blue is absorbance).(**C**) UV-visible spectrophotometer showed absorbance of folic acid in supernatant compared to totally free folic acid. (**D**) FTIR spectra of FA, PRM, and FA-PRM.

**Figure 5 ijms-24-03548-f005:**
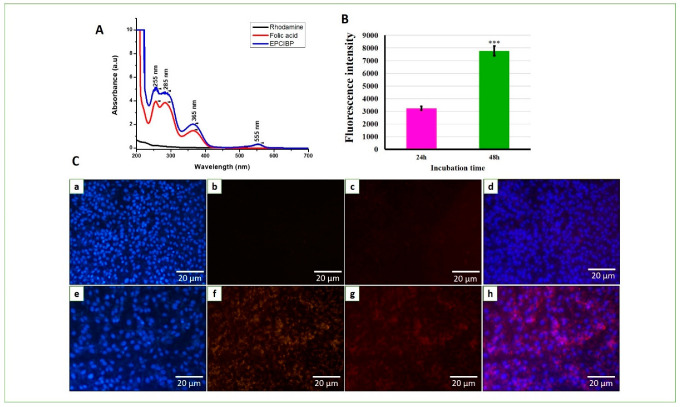
UV-visible results for R6G conjugated to EPCIBP (**A**). Fluorescence image of EPCIBP-conjugated R6G (**B**). Quantification analysis of fluorescence intensity using R6G-labeled nanoparticles under fluorescence microscopy over 24 h with *** *p* < 0.0001. (**C**). Fluorescence images demonstrate cellular internalization of EPCIBP (without FA) conjugated R6G in A549 cells; (**a**) DAPI stain to show nuclear morphology (blue color). (**b**) FITC channel. (**c**) TRIC channel of R6G (red color). (**d**) Merged images between the FITC channel and TRIC channel produced using the Image J program. Fluorescence images demonstrate cellular internalization of EPCIBP (with FA) conjugated R6G in A549. (**e**) DAPI stain to show nuclear morphology. (**f**) FITC channel. (**g**) TRIC channel of R6G. (**h**) Merged image between the FITC channel and TRIC channel produced using the Image j program.

**Figure 6 ijms-24-03548-f006:**
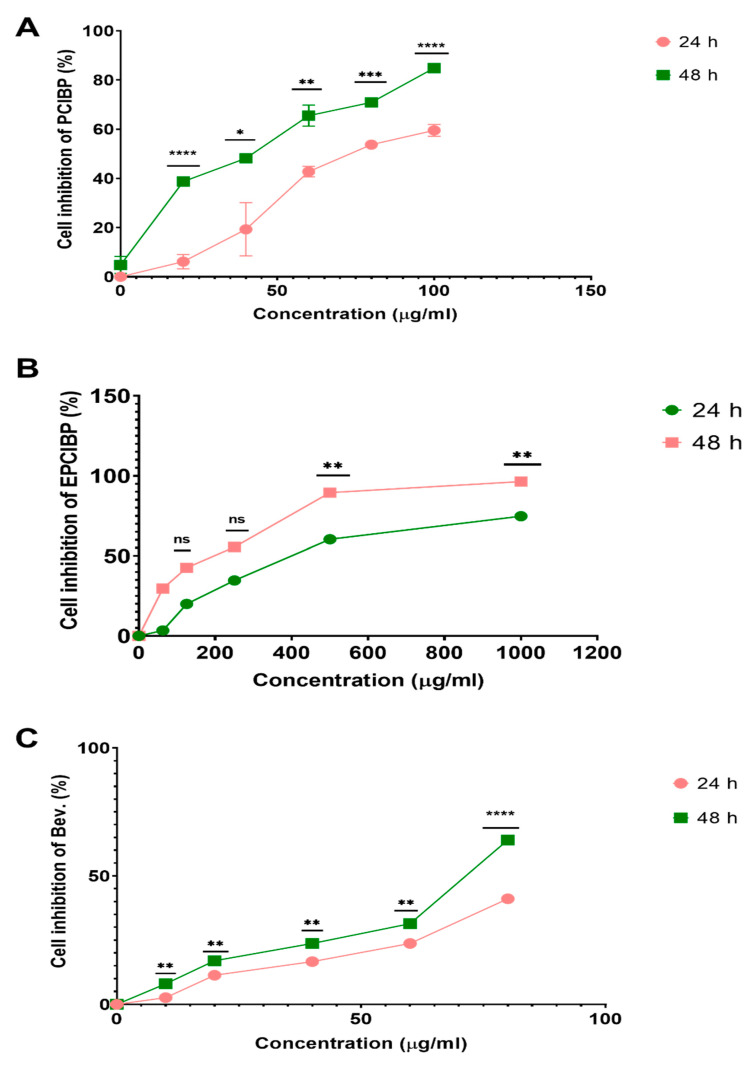
Line graph of the cell inhibition results on A549 lung cancer cells after incubation for 24 and 48 h with PCIBP, EPCIBP, and Bev. treatments. The percentage of treated cells was measured by MTT cytotoxicity assay. (**A**) PCIBP for each point compared with the other points by multiple t-tests where * *p* ≤ 0.01, ** *p* ≤ 0.001, *** *p* ≤ 0.0001, **** *p* ≤ 0.00001. (**B**) EPCIBP for each point compared with the opposite point by multiple *t*-tests where ns: nonsignificant, ** *p* ≤ 0.001, (**C**) Bev. for each point compared with the other points by multiple *t*-tests where ** *p* ≤ 0.001, **** *p* ≤ 0.00001.

**Figure 7 ijms-24-03548-f007:**
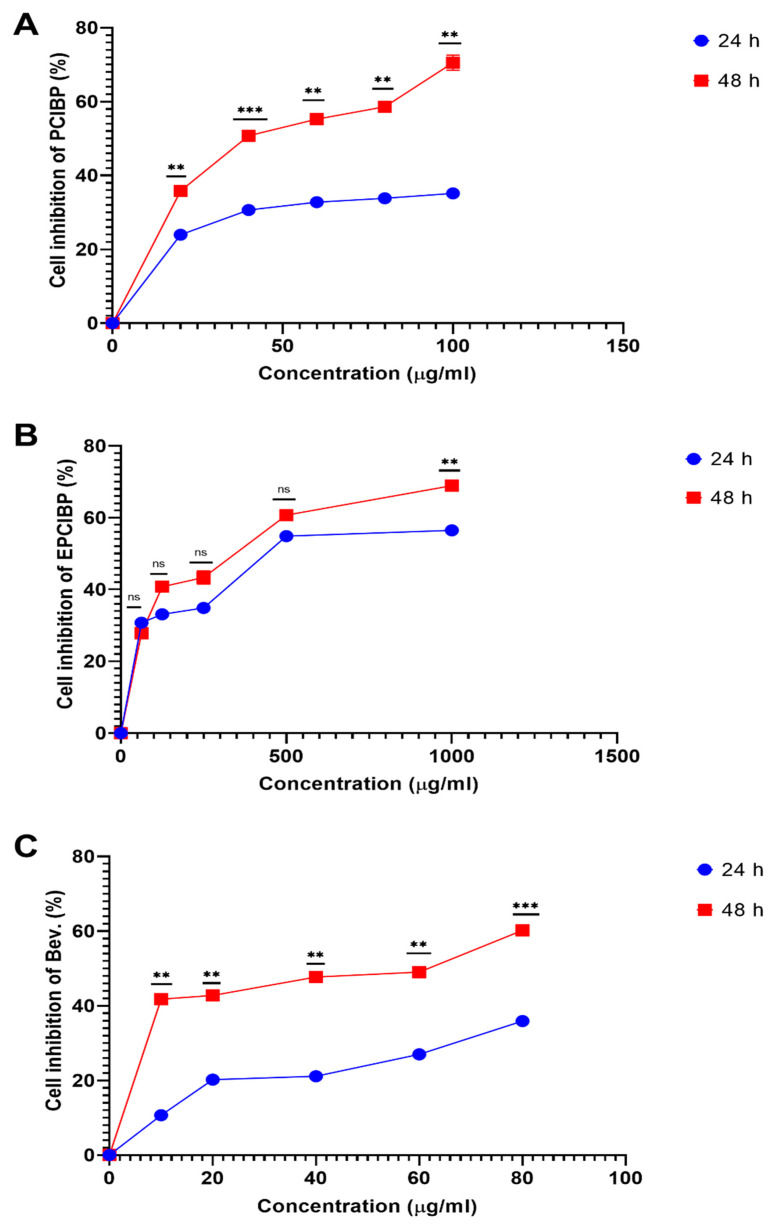
Line graph of the cell inhibition results on MCF7 breast cancer cells after incubation for 24 and 48 h with PCIBP, EPCIBP, and Bev. treatments. The percentage of treated cells was measured by MTT cytotoxicity assay. (**A**) PCIBP for each point compared with the other points by multiple *t*-tests where ** *p* ≤ 0.001, *** *p* ≤ 0.0001; (**B**) EPCIBP for each point compared with the other points by multiple *t*-tests where ns: nonsignificant, ** *p* ≤ 0.001; (**C**) Bev. for each point compared with the other points by multiple *t*-tests where ** *p* ≤ 0.001, *** *p* ≤ 0.0001.

**Figure 8 ijms-24-03548-f008:**
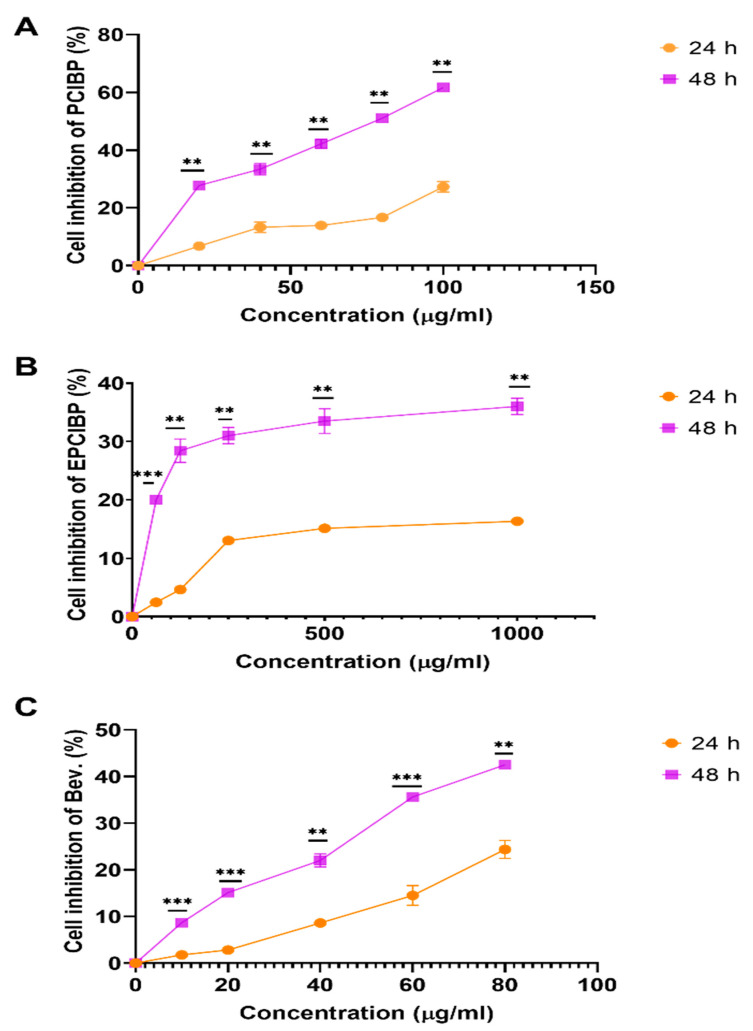
Line graph of the cell inhibition results on Vero normal cells after incubation for 24 and 48 h with PCIBP, EPCIBP, and Bev. treatments. The percentage of treated cells was measured by MTT cytotoxicity assay. (**A**) PCIBP for each point compared with the other points by multiple *t*-tests where ** *p* ≤ 0.001; (**B**) EPCIBP for each point compared with the other points by multiple *t*-tests where ** *p* ≤ 0.001, *** *p* < 0.0001; (**C**) Bev. for each point compared with the other points by multiple *t*-tests where ** *p* ≤ 0.001, *** *p* ≤ 0.0001.

**Figure 9 ijms-24-03548-f009:**
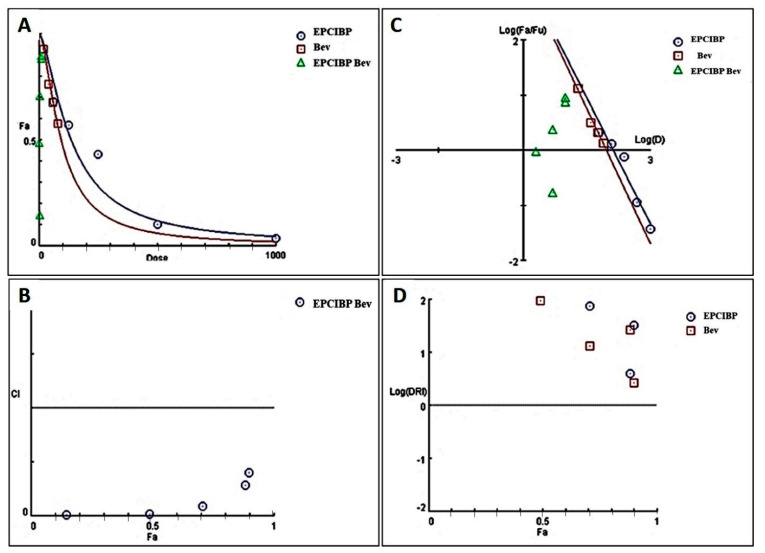
Combination index histograms. (**A**) The dose–effect curve showing combination doses on perpendicular axes that correlate to both drug levels; (**B**) the combination index plot showing an effective synergy (CI < 1) for all pairings; (**C**) plot of the treatments’ and drugs’ combined median effect levels; (**D**) plot of the non-constant combination of EPCIBP and Bev.’s log dose-reduction index (Log DRI). Fa: Default effect level.

**Figure 10 ijms-24-03548-f010:**
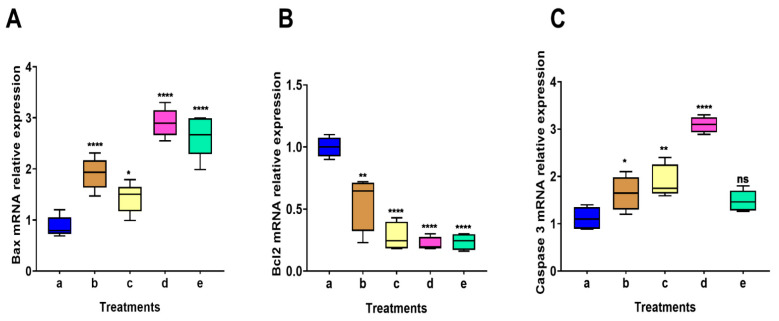
Average qRT-PCR analysis data of the mRNA expression data in different A549 treatments. a: untreated cells; b: PCIBP; c: EPCIBP; d: bevacizumab; e: EPCIBP+ Bev. combination normalized relative to the B-actin endogenous housekeeping gene. (**A**): Bax gene, * *p* ≤ 0.05, **** *p* ≤ 0.0001 vs. untreated cells, **** *p* ≤ 0.0001 vs. untreated cells. (**B**): Bcl-2, ** *p* ≤ 0.005 **** *p* ≤ 0.00001 vs. untreated cells. (**C**) Caspase 3, * *p* ≤ 0.05, ** *p* ≤ 0.005, **** *p* ≤ 0.0001, ns: nonsignificant vs. untreated cells.

**Figure 11 ijms-24-03548-f011:**
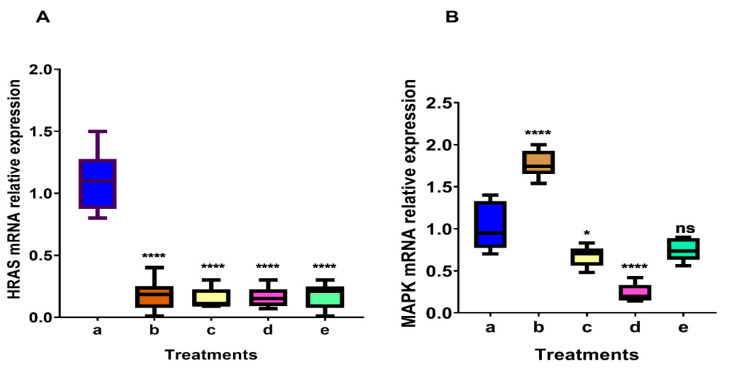
Average qRT-PCR analysis data of the mRNA expression in different A549 treatments. a: untreated cells; b: PCIBP; c: EPCIBP; d: Bev.; e: EPCIBP + Bev. combination normalized relative to the B-actin endogenous housekeeping gene. (**A**) *HRAS* gene where **** *p* ≤ 0.0001 vs. untreated cells; (**B**) *MAPK* gene where * *p* ≤ 0.02, **** *p* ≤ 0.0001, ns: nonsignificant vs. untreated cells.

**Figure 12 ijms-24-03548-f012:**
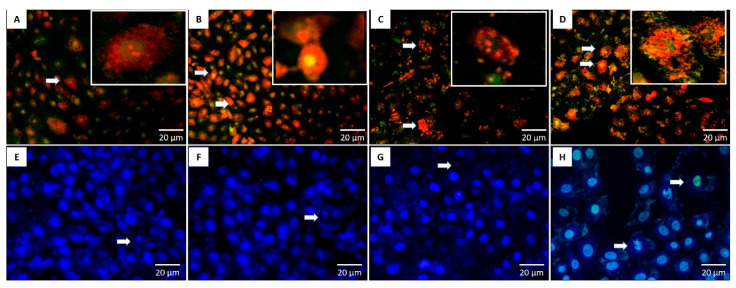
Fluorescence microscopy showing the morphology of A549 cells stained with AO/EB and DAPI after their exposure to (**A**,**E**) PCIBP, (**B**,**F**) EPCIBP, (**C**,**G**) Bev., or (**D**,**H**) a combination of EPCIBP and Bev. (arrows reveal DNA fragmentation, chromatin condensation, and cytoplasmic degradation; yellow is an early apoptotic stage. Red is a late apoptotic stage).

**Figure 13 ijms-24-03548-f013:**
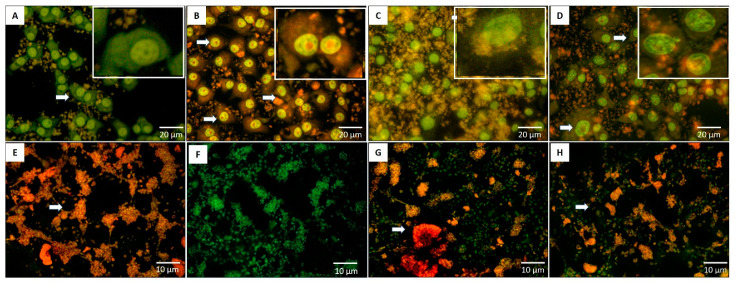
Fluorescence microscopy showing the morphology of MCF-7 cells and normal Vero cells stained with AO/EB after their exposure to (**A**,**E**) PCIBP, (**B**,**F**) EPCIBP, (**C**,**G**) Bev., or (**D**,**H**) a combination of EPCIBP and Bev. (arrows reveal early and late apoptotic stages; yellow is an early apoptotic stage. Red is a late apoptotic stage).

**Table 1 ijms-24-03548-t001:** Quantification and identification of flavonoid contents isolated from PCIBP and EPCIBP.

Flavonoids	PCIBP	EPCIBP
	Conc. (µg/mL)	Conc. (µg/mL)
Gallic acid	1.38 ± 0.06	5.59 ± 0.35
Chlorogenic acid	0.24 ± 0.02	0.13 ± 0.01
Catechin	1.42 ± 0.01	0.64 ± 0.01
Methyl gallate	0.12 ± 0.01	0.055 ± 0.001
Caffeic acid	0.09 ± 0.001	0.37 ± 0.01
Syringic acid	3.84 ± 0. 5	2.03 ± 0.02
Rutin	0.26 ± 0.01	0.18 ± 0.01
Pyro catechol	0.32 ± 0.02	0.29 ± 0.01
Coumaric acid	0.27 ± 0.01	0.14 ± 0.01
Vanillin	0.082 ± 0.004	0.083 ± 0.001
Ferulic acid	0.39 ± 0.01	0.18 ± 0.01
Cinnamic acid	3.89 ± 0.16	4.21 ± 0.19
Naringenin	0.31 ± 0.005	0.44 ± 0.01
Quercetin	6.43 ± 0.18	5.23 ± 0.01
Taxifolin	0.7 ± 0.07	2.52 ± 0.15

**Table 2 ijms-24-03548-t002:** CI data for the combined effect of Bev. and EPCIBP on A549 cells.

IC50 Dose 26.1 µg/mL Bev.	IC50 Dose 132.4 µg/mL EPCIBP	Effect (Mean Cell Viability)	CI *	Drug–Drug Interaction
1.0	1.0	0.4891	0.01771	Synergistic <1
4.0	1.0	0.7081	0.08764	Synergistic <1
1.0	4.0	0.1471	0.01305	Synergistic <1
9.0	1.0	0.8995	0.40409	Synergistic <1
1.0	9.0	0.8839	0.28923	Synergistic <1

*: CI = combination index [19].

## Data Availability

Data available in a publicly accessible repository.

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
