# Peer review of "Simultaneous Administration of Bevacizumab with Bee-Pollen Extract-Loaded Hybrid Protein Hydrogel NPs Is a Promising Targeted Strategy against Cancer Cells"

_ijms, 2023, doi:10.3390/ijms24043548_

Round 1

Reviewer 1 Report

This work is devoted to the investigation of the possibility to use encapsulated bee pollen polyphenolic compounds simultaneously with Bevacizumab for advanced non-small cell lung cancer therapy.

Although the subject of the manuscript addressed by the authors is interesting, the presentation it is not convincing. Thus, I would highlight the following points that it is required /is needed to improve and increase the interest for the paper

General comments:

- Results - brief presentation, without any comments on the obtained data;  

-Discussion section-should be improved with a more comprehensive comments related to the biological significance of the results.

-A final English spelling and grammar of the text is necessary.

Specific comments:

-Table 1-Flavonoids concentrations are different in PCIBP compared to EPICBP; this fact should be explained;

- The authors claim “The EPCIBP might hinder the release of polyphenolic compounds resulting in the keeping of total polyphenolic compounds in developed NPs compared to fresh bee pollen”, but experiments regarding the stability and amount of polyphenolic compounds released from EPCIBP (after 24, 48 hrs or longer time) are missing;

-Fig.2 Histogram of free capsule diameter should be added.

-Fig.4 A and B overlap;

-Experiments using normal cell line to check the toxicity of PCIBP, EPCIBP, Bev, or a combination of EPCIBP + Bev are missing;

-Conclusion-Egyptian (?) EPCIBP!

Based on my comments the paper needs major improvements before publication.

Author Response

Reviewer 1

Comments and Suggestions for Authors

This work is devoted to the investigation of the possibility to use encapsulated bee pollen polyphenolic compounds simultaneously with Bevacizumab for advanced non-small cell lung cancer therapy.

Although the subject of the manuscript addressed by the authors is interesting, the presentation it is not convincing. Thus, I would highlight the following points that it is required /is needed to improve and increase the interest for the paper

The authors would like to thank Reviewer #1  for his/her helpful comments that allowed us to revise our manuscript. We have made several revisions to the manuscript following your suggestions, and your concern has been carefully addressed, as can be seen below. Modifications are the green highlight words in the markedly revised manuscript to help the Reviewer and the Editor more straightforwardly check on the changes we made.

General comments:

Question 1: - Results - brief presentation, without any comments on the obtained data;  

Response to Question 1: many paragraphs were added in each presentation thoroughly over all the text to improve the presentation and to clarify the obtained data.

Question 2: -Discussion section-should be improved with a more comprehensive comments related to the biological significance of the results.

Response to Question 2: We would like to thank reviewer #1   for his/her great comment. The comment was addressed and the  discussion was improved accordingly the reviewer’s comment

Question 3: -A final English spelling and grammar of the text is necessary.

Response to Question 3: Authors ask the reviewer #1   kindly to accept our apologies. English Typos errors were captured and English is revised by ourselves because here,  we do not have an official office for English editing. 

Specific comments:

Question 4-Table 1-Flavonoids concentrations are different in PCIBP compared to EPICBP; this fact should be explained;

Response to Question 4: We would like to thank reviewer #1   so much for this valuable comment. While encapsulation of bee pollen extract resulted in minimizing the concentrations of its polyphenolic content, Gallic acid Taxifolin is still the major. This result could be explained by the fact that polyphenolic compounds (Crude) may provide various interactions during their encapsulation depending on their chemical reaction with a certain types of polymers or materials. Since polyphenolic compounds mainly contain carboxylic groups and hydroxyl groups that can react either by ester bond or amide bond. Based on their interaction and location inside capsules, they may be isolated and released from their location simply or they can be attached strongly and stayed in the inner face of polymer moieties. Thereby isolating polyphenolic compound from capsules don’t express their real concentration.

  1. da Rosa, C.G.; Borges, C.D.; Zambiazi, R.C.; Rutz, J.K.; da Luz, S.R.; Krumreich, F.D.; Benvenutti, E.V.; Nunes, M.R. Encapsulation of the Phenolic Compounds of the Blackberry (Rubus Fruticosus). LWT-Food Sci. Technol.201458, 527–533.

2.       Wang,Y., Xie,Y.,Wang,A.,Wang,J.,Wu,X., Wu,Y.,Fu,Y.,Sun,H. Insights into interactions between food polyphenols and proteins: An updated overview. Food processing and preservation.2022; 4685: e16597

Question 5- The authors claim “The EPCIBP might hinder the release of polyphenolic compounds resulting in the keeping of total polyphenolic compounds in developed NPs compared to fresh bee pollen”, but experiments regarding the stability and amount of polyphenolic compounds released from EPCIBP (after 24, 48 hrs or longer time) are missing;

Response to Question 5: We would like to provide our sorry because HPLC was not working at the moment. The experiment will be checked in our future work.

Question 6-Fig.2 Histogram of free capsule diameter should be added.

Response to Question 6: A histogram of free capsules was added and a new SEM image of free capsules was also added ( Fig2.;G-I)

Question 7-Fig.4 A and B overlap

Response to Question 7: Overlap of A and B was removed

Question 8 -Experiments using normal cell line to check the toxicity of PCIBP, EPCIBP, Bev, or a combination of EPCIBP + Bev are missing;

Response to Question 8: Normal cells were used to check the toxicity of PCIBP, EPCIBP, Bev and their combination. The result was added in (Fig. 8).

Question 9 -Conclusion-Egyptian (?) EPCIBP!

Response to Question 9: Egyptian was written here, to confirm the source of bee pollen.

Based on my comments the paper needs major improvements before publication.

Reviewer 2 Report

In the current manuscript, the authors prepared polyphenolic compounds from bee pollen (PCIBP) and encapsulated (EPCIBP) into moieties of hybrid polymeric protein hydrogels nanoparticles in which BSA was combined with protamine-free sulfate and targeted with folic acid (FA). The apoptotic effects of PCIBP and its encapsulation were verified using A549 cell lines. It was shown that the effect was synergistically improved in combination with Bev. In general, the idea presented in the paper was straight forward. However, important experiments were missing. Typically, the therapeutic effect of the developed formulation was verified using only one cell line (A549) without comparison with other cell lines. Therefore, it cannot be recommended for publication in IJMS. The following issues should be addressed:

1.       Lines 80-81: “Additionally, folic acid-conjugated protamine was used to improve the efficacy of tumor-targeted delivery”. Did the A549 cell line used in the manuscript express/overexpress folate receptors?

2.       Figure 2. It seems that EPCIBP NPs were highly aggregated so how did the author obtain the average size of 116 nm as indicated in Figure 2C? It should be much bigger than that value.

3.       Errors in figure 4. 4A was covered by 4B; 4C label should be re-arranged. In addition, in figure 4B, the column labels are missing; statistical analysis should also be performed.

4.       Texts in lines 251-253 should be referred to Figure 4A.

5.       Figure 5B. The authors should explain why at concentration of 1000 ug/ml the cell inhibition of EPCIBP treated after 24 h and 48 h was the same.

6.       Figure 5C. The authors should explain why at concentration of 80 ug/ml the cell inhibition of Bev. treatment after 24 h was higher than at 48 h. Logically, the opposite fact should be observed.

7.       In all related figures and texts: hrs units are not standard. They should be replaced by h.

8.       Figure 10. What do white arrows mean? Figure 10 was also not mentioned in the text.

9.       The therapeutic effects against different cell lines (other than A549) must be evaluated and compared. Otherwise, the authors cannot state that “…bee pollen extract loaded hybrid protein hydrogel targeted NPs is a promising strategy against non-small lung cancer cells”.

10.   Line 337. What does (106) mean?

11.   Line 397. Error in (1X104 cells/well).

12.   Line 350. Same error as in 397. 

Author Response

Reviewer 2

Comments and Suggestions for Authors

In the current manuscript, the authors prepared polyphenolic compounds from bee pollen (PCIBP) and encapsulated (EPCIBP) into moieties of hybrid polymeric protein hydrogels nanoparticles in which BSA was combined with protamine-free sulfate and targeted with folic acid (FA). The apoptotic effects of PCIBP and its encapsulation were verified using A549 cell lines. It was shown that the effect was synergistically improved in combination with Bev. In general, the idea presented in the paper was straight forward. However, important experiments were missing. Typically, the therapeutic effect of the developed formulation was verified using only one cell line (A549) without comparison with other cell lines. Therefore, it cannot be recommended for publication in IJMS. The following issues should be addressed:

The authors would like to thank Reviewer #2  for his/her helpful comments that allowed us to revise our manuscript. We have made several revisions to the manuscript following your suggestions, and your concern has been carefully addressed, as can be seen below. Modifications are the green highlight words in the markedly revised manuscript to help the Reviewer and the Editor more straightforwardly check on the changes we made.

Question 1: Lines 80-81: “Additionally, folic acid-conjugated protamine was used to improve the efficacy of tumor-targeted delivery”. Did the A549 cell line used in the manuscript express/overexpress folate receptors?

Response to Question 1: we would like to thank reviewer #2 for his/her great and valuable comment. A549 cell lines that were demonstrated for non-small lung cancer, can express folate receptors strongly on their surface. This is according to the results of immunohistochemistry, flow cytometry, and immunoassay that were published in 2018 by  (Predina et al.,2018).

  1. Predina JD, Newton AD, Connolly C, Dunbar A, Baldassari M, Deshpande C, Cantu E 3rd, Stadanlick J, Kularatne SA, Low PS, Singhal S. Identification of a Folate Receptor-Targeted Near-Infrared Molecular Contrast Agent to Localize Pulmonary Adenocarcinomas. Mol Ther. 2018 Feb 7;26(2):390-403. doi: 10.1016/j.ymthe.2017.10.016.

Question 2: Figure 2. It seems that EPCIBP NPs were highly aggregated so how did the author obtain the average size of 116 nm as indicated in Figure 2C? It should be much bigger than that value.

Response to Question 2: We would like to thank reviewer #2 for his/her comment.  The diameter of EPCIBP NPs was studied by DLS to confirm it is 138 nm. while SEM image (Fig. 2(A-C)) showed the average of diameter is 116 nm because DLS can detect the diameter of NPs in solution and the hydrodynamic behavior of NPs could be increased.

Question  3: Errors in figure 4. 4A was covered by 4B; 4C label should be re-arranged. In addition, in figure 4B, the column labels are missing; statistical analysis should also be performed.

Response to Question 3. we would like to thank reviewer #2 for his/her great comments. Figure 4 was corrected and also, and the statistical analysis was added.

Question 4: Texts in lines 251-253 should be referred to Figure 4A.

Response to Question 4. We thank reviewer #2  so much for his/her valuable comment. the figure was converted to Fig 5A according to the arrangement of the figures and the test was referred to it.

Question 5: Figure 5B. The authors should explain why at concentration of 1000 ug/ml the cell inhibition of EPCIBP treated after 24 h and 48 h was the same.

Response to Question 5. We thank reviewer #2  so much for his/her valuable comment. We would like first to express our strong apologies for this mistake.  Please accept our sorry. The result was corrected

Question 6: Figure 5C. The authors should explain why at concentration of 80 ug/ml the cell inhibition of Bev. treatment after 24 h was higher than at 48 h. Logically, the opposite fact should be observed.

Response to Question 6. We thank reviewer #2  so much for his/her valuable comment. We would like first to express our strong apologies for this mistake.  Please accept our sorry. The result was corrected.

Question 7: In all related figures and texts: hrs units are not standard. They should be replaced by h.

Response to Question 7. We thank reviewer #2  so much for his/her valuable comment. Hrs units were corrected and replaced by h.

Question 8: Figure 10. What do white arrows mean? Figure 10 was also not mentioned in the text.

Response to Question 8. We thank reviewer #2  so much for his/her valuable comment. The arrows were pointed out and Figure 10 was mentioned in the text.

Question 9: The therapeutic effects against different cell lines (other than A549) must be evaluated and compared. Otherwise, the authors cannot state that “…bee pollen extract loaded hybrid protein hydrogel targeted NPs is a promising strategy against non-small lung cancer cells”.

Response to Question 9. We thank reviewer #2  so much for his/her valuable comment.. New experiments were added to investigate the cytotoxicity of PCIBP, EPCIBP, Bev and their combination in normal cell line (Vero) and breast cancer cell line (MCF-7). The result was added in Figures 7 and 8. Additionally, Acridine orange/ Ethidium bromide was used  to demonstrate the cell morphology in normal cells and MCF-7 (Fig 13).

This sentence “…bee pollen extract loaded hybrid protein hydrogel targeted NPs is a promising strategy against non-small lung cancer cells”  was changed into  “…bee pollen extract loaded hybrid protein hydrogel targeted NPs is a promising strategy against cancer cells”

Question 10: Line 337. What does (106) mean?

Response to Question 9. We thank reviewer #2  so much for his/her valuable comment

It was corrected to be 106.

Question 11: Line 397. Error in (1X104 cells/well).

Response to Question 9. We thank reviewer #2  so much for his/her valuable comment

It was corrected to be 1x104.

Question 12: Line 350. Same error as in 397. 

Response to Question 9. We thank reviewer #2  so much for his/her valuable comment

It was corrected to be 1x104..

Round 2

Reviewer 1 Report

The manuscript entitle “Simultaneous administration of bevacizumab with bee pollen extract loaded hybrid protein hydrogel targeted NPs is a promising strategy against cancer cells” was revised and the version 2 revealed an improved form.

I have only one suggestion:

- A final English spelling and grammar of the text!

In conclusion, the manuscript could be published in the presented revised form.

Reviewer 2 Report

The authors have addressed all issued pointed out the Reviewer. The quality of the manuscript has been significantly improved. Therefore, it is recommended for publication in the journal.